# LEARNING CONDITIONAL POLICIES FOR CRYSTAL DESIGN USING OFFLINE REINFORCEMENT LEARNING

## ABSTRACT

Navigating through the exponentially large chemical space to search for desirable materials is an extremely challenging task in material discovery. Recent developments in generative and geometric deep learning have shown promising results in molecule and material discovery but often lack evaluation with high-accuracy computational methods. This work aims to design novel and stable crystalline materials conditioned on a desired band gap. To achieve conditional generation, we: 1. Formulate crystal design as a sequential decision-making problem, create relevant trajectories based on high-quality materials data and use conservative Q-learning to learn a conditional policy from these trajectories. To do so, we formulate a reward function that incorporates constraints for energetic and electronic properties obtained directly from density functional theory (DFT) calculations; 2. Evaluate the generated materials from the policy using DFT calculations for both energy and band gap; 3. Compare our results to relevant baselines, including a random policy, behavioral cloning, and unconditioned policy learning. Our experiments show that our conditioned policies achieve more targeted crystal structure designs and demonstrate the capability to perform crystal structure design evaluated with accurate and computationally expensive DFT calculations.

## 1 INTRODUCTION

The widespread enthusiasm in exploiting artificial intelligence (AI) for scientific discovery (Wang et al., 2023) has resulted in various methodologies to integrate existing scientific knowledge and large databases to design and test new hypotheses more quickly. Recently, AI has shown favorable results in expediting the discovery of new chemical structural entities (e.g., small molecules, materials, and polymers) (Jain et al., 2023; Xu et al., 2023; Bran et al., 2023; Sim et al., 2023). While several studies have focused on small molecule design for applications in drug discovery, there has also been an upsurge in attention for AI-based material discovery (Miret et al., 2022b; Song et al., 2023; Lee et al., 2023; Miret et al., 2022a). Among solid-state materials, crystalline substances are abundant in nature and are extensively used in industry for designing batteries, semiconductors and photovoltaic systems. The set of known and experimentally observed crystalline materials is an infinitesimally tiny fraction (around 200,000) of the exponentially large chemical space spanning over 100 elements in the periodic table and 230 space groups in 3 dimensions (Rutherford, 2005; Zhao et al., 2023). Determining a way to navigate through this large space to select chemical candidates with desired properties would be immensely beneficial for a plethora of applications like designing energy-efficient semiconductors and combatting climate change.

Besides the complex nature of the chemical space, designing stable crystalline materials using computational chemistry is a long-standing challenge primarily due to the time-consuming density functional theory (DFT) calculations to estimate energetic and electronic properties of materials. Previous works have utilized generative adversarial networks (GANs) (Nouira et al., 2018), diffusion models (Xie et al., 2021), and reinforcement learning (RL) (Meldgaard et al., 2020), in addition to advanced crystal representation schemes for generating crystals (Damewood et al., 2023; Duval et al., 2022). However, we identify two major gaps in the existing literature for AI-based material discovery. Firstly, most methods do not incorporate quantum mechanics-based first-principles calculations in the learning model, and instead use ML approximators. Studies that incorporate DFT computations in their ML pipeline for material design usually focus on smaller and very specific chemical systems (with limited number of elements or constraints on the space group) that might

not generalize well to diverse chemical systems (Meldgaard et al., 2020; Zhao et al., 2021). Secondly, state-of-the-art generative AI methods, such as diffusion models, predict the identities and positions of all atoms simultaneously, which is orthogonal to sequence based RL methods that also have more established exploration methods applicable to vast search spaces.

In this work, we further the state-of-the-art in the crystal design problem by developing a model which learns to sequentially construct crystal skeleton graphs by optimizing for both lower total energy and desired band gap value (energy gap between the valence and conduction bands in solids), as computed by DFT. In our case, the crystal lattice parameters and positions of atomic sites are known beforehand (crystal skeleton) and the task is to learn a conditional policy that can sequentially fill atoms to generate a stable and valid crystal with a desired band gap energy. To alleviate the issue of time-consuming DFT calculations when integrated in the scientific discovery loop, we apply offline reinforcement learning using the conservative Q-learning (CQL) approach (Kumar et al., 2020), which is known to mitigate overestimation and out-of-distribution issues when agents are trained with static datasets in an offline manner. We construct a state transition dataset from high-quality nonmetallic crystal structures present in the Materials Project database (Jain et al., 2013). Moreover, we augment this dataset to reduce the order dependence of nodes while training our offline policy. The reward function is carefully formulated to penalize high energies and large deviations from the desired band gap. Further, we leverage an expressive graph neural network (GNN) for crystal representation that ensures invariance to periodicity, translation, and rotation. Through our work, we aim to accelerate the process of high-throughput virtual screening (HTVS) for materials, where usually elements are combinatorially substituted in a known crystal structure and optimized using DFT calculations. Overall, our contributions are three-fold, as follows:

1. **DFT Evaluation of Crystal Structures with Reinforcement Learning**: Our distinct formulation of the reward function for offline RL is crafted from total energy and band gap values computed using first principles DFT calculations, the gold standard of computational chemistry. The reward function penalizes high total energy and large deviations from the desired band gap to a policy conditioned on a targeted band gap value.

2. **Conservative Offline Reinforcement Learning Approach:** Using CQL as our offline RL framework, we demonstrate that conservatism, combined with the right amount of importance for the energy and band gap terms in the reward function, can lead the agent to generate crystals with a favorable shift in the distribution of properties of interest. This achievement is noteworthy, especially considering our task has a very sparse reward scheme, allows no exploration, and has a high dimensional action space and limited data.

3. **Open-Source Crystal Structure Design Trajectory Data:** To ensure consistency in our reward calculation, we evaluate 20k crystal structures using the Quantum Espresso (Giannozzi et al., 2009) package for DFT calculation and subsequently construct offline RL trajectories based on the data. We plan to release the dataset of trajectories and calculations as part of the paper to enable research to further improve our work. The release of the data is noteworthy, given that we use an open-source DFT calculator that is highly reproducible and consistent for all the structures evaluated. Prior work used different types of proprietary DFT software, which is difficult for the research community to reproduce.

## 2 RELATED WORKS

**Automated Materials Design.** Prior work has explored the application of various types of methods to crystal structure design, including evolutionary algorithms, simulated annealing, particle swarm optimization, and high-throughput screening (Glass et al., 2006; Doll et al., 2008; Wang et al., 2012). Machine learning based methods have been more recently applied, primarily to molecular design problems, but also to periodic crystal structures (Li et al., 2020; Damewood et al., 2023). Moreover, there have been notable works using machine learning based methods to approximate the evaluation of material properties and behaviors (Miret et al., 2023; Lee et al., 2023). This includes approximating DFT outputs directly for different systems, such as ground-state crystal structures for a variety of applications, such as catalysts (Chanussot et al., 2021; Chen & Ong, 2022). The recent progress in graph neural networks and generative models have led to their successful application in materials design (Duval et al., 2023; Chen & Ong, 2022). GANs have been well explored for crystal structure design (Nouira et al., 2018; Zhao et al., 2021; Kim et al., 2020). However, these approaches restrict the complexity of the problem to a fixed crystal system or a smaller chem-

ical space . Zhao et al. (2023) proposed a physics-guided GAN model using convolutional layers to learn the generative distribution of stable crystals, and the evaluation of generated crystals was done using DFT. CDVAE (Xie et al., 2021) introduced a diffusion-based framework with highly expressive graph representation learning techniques to generate stable and valid crystal structures in 3 dimensions. Zheng et al. (2023) used their Distributional Graphormer to generate structures of carbon polymorphs with the desired band gap. Meldgaard et al. (2020) focused on building an online RL framework with DFT integrated reward function for surface reconstructions. However, they use the tight-binding version of DFT (DFTB), whose accuracy is lower than full DFT calculations. Other relevant works include Pan et al. (2022); Sui et al. (2021); Law et al. (2022) and Zheng et al. (2022).

**Offline Reinforcement Learning.**    Offline RL (Levine et al., 2020; Prudencio et al., 2023) enables for learning an optimal policy directly from trajectories, making it possible to utilize knowledge from existing crystal structures. The ability to learn from previously determined crystal structures reduces the need for costly DFT calculations during training which are necessary for online RL methods. Many recently proposed offline RL methods focus on managing distribution shift between the offline data and the learned policy (Nair et al., 2021; Kostrikov et al., 2022; Yu et al., 2021), with Conservative Q-Learning (CQL) (Kumar et al., 2020) proving to be a particularly robust approach. CQL has shown success in training large capacity models and performing better with suboptimal data, which makes it a particularly good fit for our crystal structure design case.

## 3    BACKGROUND

### 3.1    CRYSTALS

Solid-state crystals are characterized by ordered and periodic arrangement of atoms in 3 dimensional space. They consist of unit cells, which are the smallest group of atoms that form the repeating pattern of the crystal. A crystal's composition and arrangement of atoms gives rise to distinct electronic properties usually determined by experimental or simulation-based density functional theory (DFT) calculations. In 3 dimensions, we can mathematically express the unit cell $U$ as follows.

$$U = \left\{ w_1 l_1 + w_2 l_2 + w_3 l_3 \mid 0 \leq w_i < 1 \right\}, \tag{1}$$

where $l_1, l_2, l_3 \in \mathbb{R}^3$ are primitive translation vectors that define the periodic translation symmetry of the crystal. Discrete linear transformations can be performed to obtain unit cells at different locations with $\nabla = c_1 l_1 + c_2 l_2 + c_3 l_3$, where $c_1, c_2$, and $c_3$ are integers, thus generating the entire 3-dimensional lattice. Therefore, a 3-dimensional lattice $\Lambda$ is defined as all integral combinations of lattice basis vectors

$$\Lambda = \left\{ c_1 l_1 + c_2 l_2 + c_3 l_3 \mid c_i \in \mathbb{Z} \right\}. \tag{2}$$

For a crystal with $N$ atoms, where the atom positions are given by $X = \{x_0, \cdots, x_{N-1}\}$, the corresponding position of atom $i$ in a unit cell translated by $c_1 l_1 + c_2 l_2 + c_3 l_3$ is given by

$$x_i' = x_i + c_1 l_1 + c_2 l_2 + c_3 l_3 \tag{3}$$

Further, there are 230 space groups in the 3-dimensional space, each of which describes a specific crystal symmetry. Every crystal in the database is associated with one space group number (1–230) depending on the arrangement of atoms in the crystal lattice. The order is based on the increasing complexity of symmetry elements and their combinations. For instance, space group number 1 is the simplest and least symmetric crystal system (triclinic), and 230 has the highest degree of symmetry (cubic).

### 3.2    CRYSTAL REPRESENTATION

A natural way to represent crystals is using graphs, with atoms as nodes and edges that connect neighboring or bonded atoms. However, using simple graphs is often not expressive enough to incorporate the inherent periodicity in crystals. In this work, we adopt multigraphs, following Xie & Grossman (2018) to represent crystals structures. In multigraphs, two nodes can be connected by more than one type of edge. In the context of crystals, consider a graph $\mathcal{G} = (V, E)$ with

nodes (atoms) $V = \{v_0, \cdots, v_{N-1}\}$ and edges (neighboring atoms) $E = \{e_{uv,(c_1,c_2,c_3)} | 0 \leq u \leq N-1, 0 \leq v \leq N-1, c_1, c_2, c_3 \in \mathbb{Z}, u, v \in V\}$. Here, $e_{uv,(c_1,c_2,c_3)}$ is a directed edge from atom $u$ to atom $v$ in a unit cell translated by $c_1 \boldsymbol{l_1} + c_2 \boldsymbol{l_2} + c_3 \boldsymbol{l_3}$. If $c_1 = c_2 = c_3 = 0$, it corresponds to an edge between $u$ and $v$ in the same unit cell. Likewise, if $c_1 = 1, c_2 = c_3 = 0$ it corresponds to an edge between atom $u$ in the original unit cell and atom $v$ in a unit cell translated by $\boldsymbol{l_1}$. This way, multigraphs carry information about the entire 3 dimensional structures of crystals.

### 3.3 OFFLINE REINFORCEMENT LEARNING

While online RL methods demand frequent agent-environment interactions, offline RL exploits existing data (Prudencio et al., 2023), which is useful when receiving rewards or feedback from the environment is computationally expensive or physically implausible. As previously mentioned, our reward formulation depends on the energies and band gaps of crystals computed by DFT. Given that the time it takes for performing DFT simulation ranges between 6 seconds to more than 20 minutes for each input depending on its size and type, it is highly infeasible to train an online reinforcement learning algorithm for this problem. Additionally, the high dimensional action space and the extremely complex reward landscape with narrow modes demands large amounts of exploration while learning in an online manner. Offline RL aims to learn from a static dataset $\mathcal{D}$ consisting of state transitions, i.e., $(\boldsymbol{s_t}, \boldsymbol{a_t}, \boldsymbol{s_{t+1}}, r_t)$ obtained from a behavioral policy $\pi_\beta(\boldsymbol{a}|\boldsymbol{s})$ to learn an offline policy $\pi_o(\boldsymbol{a}|\boldsymbol{s})$. However, directly adopting model-free RL (e.g., deep Q learning) approaches in a data-driven manner causes two major issues – 1) the learned policy becomes out-of-distribution from the behavioral policy and 2) values of some states are over estimated. Both these issues go hand-in-hand. Addressing these issues, Kumar et al. (2020) proposed conservative Q-learning (CQL), which regularizes Q-values by concurrently optimizing for the Bellman error to learn a conservative and lower-bound Q function. The optimization objective of the DQN (Mnih et al., 2015) version (discrete action space) of CQL is given below

$$\min_\theta \omega \mathbb{E}_{\boldsymbol{s} \sim \mathcal{D}} \Big[ \log \sum_{\boldsymbol{a'}} \exp(Q_\theta(\boldsymbol{s}, \boldsymbol{a'})) - \mathbb{E}_{\boldsymbol{s},\boldsymbol{a} \sim \mathcal{D}} \big[ Q_\theta(\boldsymbol{s}, \boldsymbol{a}) \big] \Big] + \tag{4}$$

$$\frac{1}{2} \mathbb{E}_{\boldsymbol{s},\boldsymbol{a},\boldsymbol{s'},r \sim \mathcal{D}} \Big[ Q_\theta(\boldsymbol{s}, \boldsymbol{a}) - \Big( r + \gamma \max_{\boldsymbol{a'}} Q_{\theta'}(\boldsymbol{s'}, \boldsymbol{a'}) \Big) \Big]^2.$$

Here, $\omega$ controls the amount of conservatism, i.e., higher the value of $\omega$, the more the preference for a conservative policy that better fits the data. $Q_{\theta'}$ is the target network. When the action space is discrete, learned discrete offline policy is therefore

$$\pi_o(\boldsymbol{a}|\boldsymbol{s}) = \arg\max_{\boldsymbol{a}} Q_\theta(\boldsymbol{s}, \boldsymbol{a}). \tag{5}$$

### 3.4 DENSITY FUNCTIONAL THEORY

DFT is a simulation-based quantum mechanical modeling method that is used to compute the electronic structure of multi-atom systems, thereby estimating several properties including total energy, formation energy, and band gap. This is achieved by iteratively solving the Kohn–Sham equations (Kurth et al., 2005). For evaluating crystal structures, we make use of the open-source Quantum Espresso software suite (Giannozzi et al., 2009) to perform self-consistent field (SCF calculations) using the Perdew–Burke-Ernzerhof (PBE) exchange-correlation functional. However, the PBE functional is known for its systematic underestimation of band gap energies (Seidl et al., 1996), and is less accurate than functionals like HSE06 (Heyd et al., 2003) or other self-energy approximations like GW (Aryasetiawan & Gunnarsson, 1998). Nevertheless, we used PBE because of its lower computational costs and superiority over DFTB. The output produced by the DFT simulation consists of two important properties that we are interested in – total energy (in Rydberg) and band gap (in eV). In this process, we also faced multiple new crystals failing to complete DFT simulation due to unknown properties (e.g., spin, magnetization) as part of our evaluation.

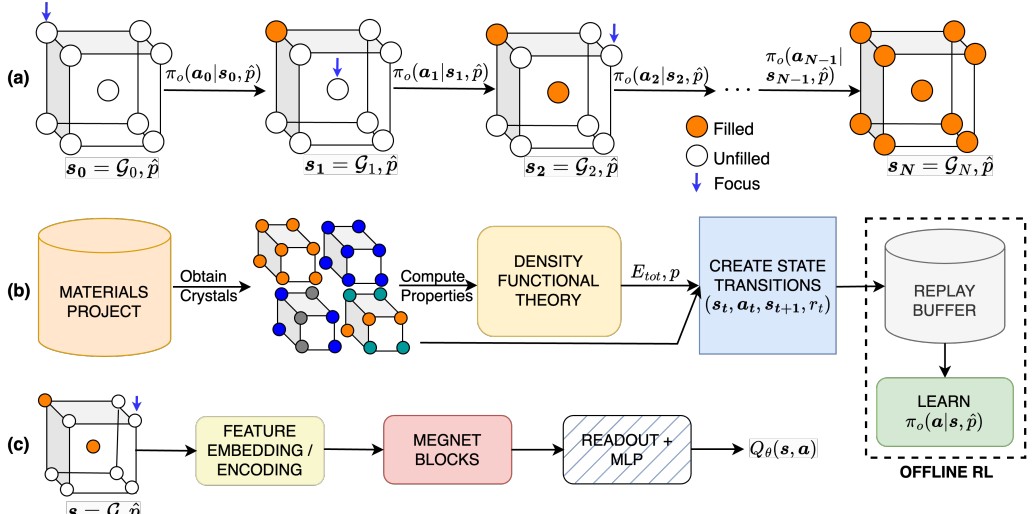

Figure 1: a) Our design approach centers on filling in the composition of predefined crystal using an RL policy. b) To successfully train an RL policy, we obtain data from Materials Project (Jain et al., 2013), recompute relevant property values using open-source DFT (Quantum Espresso (Giannozzi et al., 2009) and create trajectories for offline RL. c) We train a graph neueral network based policy based on MEGNet (Chen et al., 2019) to achieve property conditioned crystal generation.

## 4 METHODS

### 4.1 RL FORMULATION

The RL formulation of our problem follows a MDP defined as $M = \langle \mathcal{S}, \mathcal{A}, \mathcal{T}, R, \gamma \rangle$, where $\mathcal{S}$ denotes the state space, $\mathcal{A}$ denotes the action space, $\mathcal{T}(\boldsymbol{s'}|\boldsymbol{s},\boldsymbol{a}) : \mathcal{S} \times \mathcal{S} \times \mathcal{A} \rightarrow [0,1]$ is the environment transition probability function, $R(\boldsymbol{s},\boldsymbol{a}) : \mathcal{S} \times \mathcal{A} \rightarrow \mathbb{R}$ is the reward function, and $\gamma \in [0,1]$ is a discount factor denoting the preference for long term rewards over short term rewards. In our setup, the state space consists of empty, partially or fully filled multigraphs ($\mathcal{G}(V,E)$) of crystal structures. The action space $\mathcal{A}$ consists of atomic elements from which the agent has to choose to assign an atom at a given atomic site in a unit cell. Starting with initial state $\boldsymbol{s_0}$, which is the graph $\mathcal{G}_0$ of an empty crystal skeleton, the sequential construction of a crystal of $N$ atoms can be represented as a trajectory, as shown in Figure 1.

#### 4.1.1 REWARD FUNCTION

For this property-driven crystal design problem, our reward function is expected to penalize high total energies ($E_{tot}$) and large deviations from a desired property of interest (e.g., band gap), whose value is denoted by $\hat{p}$. In the context of training an offline RL agent with batches of transitions, we aim to minimize the deviation between the ground truth property $p$ of the crystal and $\hat{p}$ (desired property). This bi-objective optimization can be addressed by using a linear combination of terms that individually optimize for lower energy and desired property. In other words, for a crystal with $N$ atoms, the terminal reward, which is also equal to the return in this case, can be expressed in terms of its total energy $E_{tot}$ and ground truth property $p$ as follows.

$$r_N(E_{tot}, \hat{p}, p) = \alpha_1 g_E(E_{tot}) + \alpha_2 g_p(p, \hat{p}). \tag{6}$$

Here, $g_E(E_{tot})$ enforces lower total energy, $g_p(p, \hat{p})$ enforces $p$ and $\hat{p}$ to be close, and $\alpha_1$ and $\alpha_2$ are design parameters that control the importance of each of the terms. We emphasize that this formulation of the reward function is only reasonable when the magnitudes of $g_E(E_{tot})$ and $g_p(p, \hat{p})$ are comparable. However, because of the large discrepancy in the magnitudes of the of the total energy (Rydberg units) and the band gap (eV units), we devise $g_E(E_{tot})$ such that the energy term is scaled down to lower magnitudes, and propose an appropriate distance function for $g_p(p, \hat{p})$ in the range. To achieve this, we perform log-scaling of the total energy value, and apply an exponential

distance function to penalize large deviations from the desired property yielding:

$$r_N = \alpha_1 \log_{10}(-E_{tot}) + \alpha_2 \exp\left[-\frac{(p-\hat{p})^2}{\beta}\right].$$

(7)

This introduces another design parameter $\beta$, which essentially influences the sharpness of the mode of reward surface, with a lower value of $\beta$ resulting in a higher level of sharpness (Jain et al., 2023).

## 4.2 Q-NETWORK AND STATE REPRESENTATION

Our conditional Q-network $Q_\theta(s, a; \hat{p})$ consists of two components: 1) a graph neural network that extracts meaningful state representation of the input multigraph; 2) linear layers for computing Q-values from this representation. To represent and process multigraphs in an expressive manner, we adopt the MEGNet model Chen et al. (2019), a universal graph machine learning framework for molecules and materials. MEGNet provides an effective way of iterative information exchange among node, edge and state features, which is particularly useful for chemical entities. For a crystal graph $\mathcal{G}(V, E, y; \hat{p})$ conditioned on the desired property $\hat{p}$, $V$ and $E$ are sets of nodes and edges, and $y$ corresponds to the global state-level feature. For the N atoms in a unit cell, the categorical feature of the nodes $H = \{h_u\}_{u=0}^{N-1}$ denote the one-hot encoding of the atom type in each of the nodes. It includes an additional dimension to indicate whether the node is currently filled or unfilled with an atom. Edges connect neighboring atoms based on the CrystalNN scheme proposed by Pan et al. (2021) for determining the presence and type (i.e., $(c_1, c_2, c_3)$ triplet) of edges. The set of edge features $\mathcal{T} = \{t_{uv,(c_1,c_2,c_3)}\}$ represents the Gaussian distance between the position of atom $u$ in the reference unit cell and atom $v$ in a unit cell shifted by $c_1 l_1 + c_2 l_2 + c_3 l_3$.

$$t_{uv,(c_1,c_2,c_3)} = \exp\left[-\frac{d_{uv,(c_1,c_2,c_3)}^2}{\rho}\right],$$

(8)

$$d_{uv,(c_1,c_2,c_3)} = \sqrt{(x_v + c_1 l_1 + c_2 l_2 + c_3 l_3 - x_u)^2},$$

(9)

where $x_u, x_v \in \mathbb{R}^3$ are the positions (Cartesian coordinates) of atoms $u$ and $v$ in the reference unit cell. The state-level feature $y$ is expressed as follows.

$$y = [z || f], z = [a, b, c, \phi_1, \phi_2, \phi_3, S, \hat{p}].$$

(10)

where, $a, b, c$ are the lengths of the edges of the lattice ($a = \|l_1\|, b = \|l_2\|, c = \|l_3\|$), $\phi_1, \phi_2, \phi_3$ are the angles of the lattice, S is the space group number of the crystal, $\hat{p}$ is the desired property that the policy is conditioned on, and $f$ is a categorical feature, which we refer to as *focus* – it instructs the policy which unfilled node to focus on for atom type prediction in the following step. The categorical features $H$ and $f$ are passed through embedding layers to obtain embedded feature maps $\tilde{H}, \tilde{f}$. Numerical features $\mathcal{T}$ and $y$ are passed through multilayer perceptrons (MLPs).

$$\tilde{y} = MLP([z || \tilde{f}]).$$

(11)

A graph $\tilde{\mathcal{G}}$ with embedded and encoded features is then passed through $K$ MEGNet layers, followed by a readout layer (Appendix A.2) to obtain a graph-level representation, which is then passed through an MLP to obtain conditioned Q-values for all actions in $\mathcal{A}$.

$$\tilde{\mathcal{G}}^{(k+1)} = MEGNET(\tilde{\mathcal{G}}^{(k)}) \,\forall\, k = 0, \cdots, K-1$$

(12)

$$\psi(\tilde{\mathcal{G}}^{(K)}) = READOUT(\tilde{\mathcal{G}}^{(K)})$$

(13)

$$Q_\theta(s = \mathcal{G}; \hat{p}) = MLP(\psi(\tilde{\mathcal{G}}^{(K)}))$$

(14)

## 4.3 DATASET

For this study, we used a subset of the Materials Project database, referred to as MP-20, that was previously used by Xie et al. (2021). MP-20 consists of $\sim 45k$ metallic and nonmetallic crystals with different structure and composition, covering 88 elements in the periodic table. All of them have at most 20 atoms. For our experiments, we excluded metallic crystals with zero band gap[1]. Metals

---

[1]Metallic crystals, being conductors have a zero band gap because of the overlapping conduction and valence bands.

constituted more than 60% of the data, leading to class imbalance challenges while conditioning the model with a nonzero band gap. Next, we used Quantum Espresso to determine the total energies and band gaps of all nonmetallic crystals in the training and validation set. In the end, our training set included 8832 crystals, and our validation set included 2486 crystals.

### 4.4 STATE TRANSITIONS FOR OFFLINE RL

As shown in Figure 1, we generated a static dataset for training the offline policy using episodic trajectories consisting of $(s_t, a_t, s_{t+1}, r_t)$ transitions from MP-20 crystals. We applied a deterministic policy $\pi_\beta(a|s)$, where the actions correspond to the original element identities of the atom at a specific position of interest in an empty or partially constructed crystal skeleton graph. Each trajectory of an episode starts with the initial state $s_0$, which is a graph $\mathcal{G}_0$ of a crystal skeleton, where all atom identities are hidden. Through the *focus* feature $f$, we are explicitly providing the order of traversal through the nodes of the graph, thereby simplifying the problem further. To mitigate the effects of bias due to this order dependency, we obtain up to 5 trajectories for each crystal by varying the order of nodes with breadth-first traversals of the graph from different source nodes. This way, we obtained $\sim 520k$ transitions to train our offline RL policies.

## 5 EXPERIMENTS

In this study, we focus on designing stable (i.e., low energy) crystals that have a desired band gap ($\hat{p}$) of 1.12 eV, 2 eV, 3eV, and 4 eV, which fall within the semiconductor range. To determine the amount of conservatism required for better performance, we varied the CQL $\omega$ term using weight of 1 and 5, with the latter being more conservative than the former. Furthermore, we investigate the effect of the design parameters of the reward function in Equation (7) (i.e., coefficients $\alpha_1, \alpha_2, \beta$) on generating favorable crystals. After an initial hyperparameter sweep, we choose the coefficents as follows: $\alpha_1 = \{0, 1\}, \alpha_2 = \{5, 10\}, \beta = \{1, 3\}$. As such, we trained 16 models for each target band gap. Our baselines are 1) **Random Policy**, 2) **Behavioral Cloning** (BC)[2], and 3) **Unconditional CQL Policy** (where $\hat{p}$ is removed in the state feature vector and the reward is only in terms of $E_{tot}$). For evaluating the model, we start with an empty crystal skeleton graph $\mathcal{G}_0$ as the initial state $s_0$, and perform a rollout using the learned conditional offline policy $\pi_o(a|s, \hat{p})$ to sequentially fill atoms in the crystal. We then perform a pre-simulation assessment of the generated crystals using the following metrics – 1) **Compositional Validity**: a generated crystal is valid if it has an overall neutral charge, as computed by SMACT (Davies et al., 2019), 2) **Accuracy**, which is the fraction of correctly predicted atoms, and 3) **Similarity**, which measures the similarity of the predicted atoms with the ground truth, i.e., two atoms are similar if they belong to the same class of elements[3]. Our results are shown in Table 1 for 1.12 eV and Table 2 for 4 eV.

Next, we performed DFT simulation for all the valid crystals to estimate the total energy and band gap. The post-simulation metrics are 1) **Earth Mover Distance (EMD) between the generated and true band gap distributions** ($\Gamma_{true}^p$), 2) **Earth Mover Distance between the generated and true total energy distributions** ($\Gamma_{true}^E$), 3) % of crystals that have the band gap value in the **desired range** ($\nu$), which in our case is from $\hat{p} - 0.25$ eV to $\hat{p} + 0.25$ eV, and 4) **Out-of-distribution design** ($\kappa$) – % of generated crystals that have band gaps in the desired range but whose corresponding ground truth crystals do not have band gaps in the desired range. The results are shown in Figure 2.

### 5.1 ANALYSIS OF PRE-SIMULATION METRICS

For all band gap targets, as seen in Table 1 (for 1.12 eV) and Table 2 (for 4 eV), the more conservative model (i.e., $\omega = 5$) generally performs better in terms pre-simulation metrics. The metrics were also influenced by the magnitude of the reward function – the higher the magnitude, lower the accuracy, and in most cases, the lower the validity of generated structures. This is interesting because when the magnitude of the reward is lower or $\omega$ is higher, the conservative term in the CQL objective in Equation (4) becomes dominant, resulting in the net maximization of Q-values of state-action pairs present in the dataset. Evidently, behavioral cloning (BC), being the most conservative

---

[2]Trained with supervised classification loss

[3]Classes – transition metals, post-transition metals, group 1 metals, group 2 metals, nonmetals, lanthanides, actinides, halogens, and noble elements

approach with no reward signal, performed the best for all pre-simulation metrics, which can be attributed to BC's better prediction capacities attributed to supervised learning. However, this might not be helpful from the perspective of property-driven crystal design where the CQL-based policies outfperform BC in $\kappa$, as described next in Section 5.2 outlining relevant case studies.

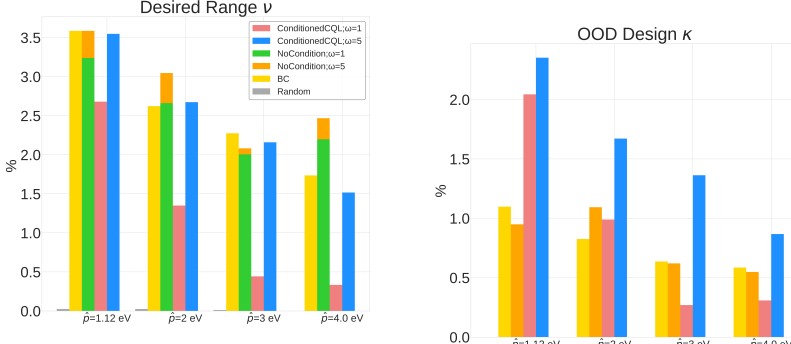

(a) % Desired range for different band gaps targets for various policies. Conditioned policies outperform random policy and compete with unconditional policies in designing crystal in the desired property range.

(b) % of generated crystals with property in the desired range with corresponding ground truth crystals outside the desired range.

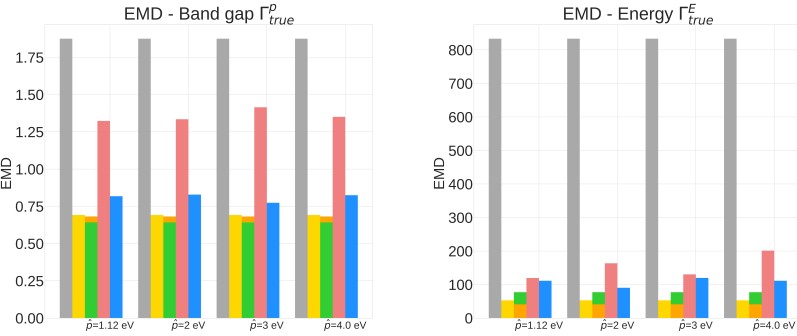

(c) Band gap EMD (generated vs true) for various policies showing that unconditioned policies reproduce the original dataset better. Lower is better.

(d) Energy EMD (generated vs true) for various policies showing that unconditioned policies reproduce the original dataset better. Lower is better.

Figure 2: Results for conditioned CQL policies on all band gap design targets. Conditioned and more conservative policies perform better in the $\kappa$ metric while unconditioned policies, including behavioral cloning, perform better at reproducing the original distribution. Random policies fail to reproduce the original distribution and achieve desired properties.

## 5.2 BAND GAP DESIGN CASE STUDIES: TARGETING 1.12 EV, 2 EV, 3 EV & 4 EV

The results in Figure 2, which include a well-performing policy for all the design cases, show some clear trends: 1) Conditioned policies (with $\omega = 5$) generate more materials in the desired property range when the corresponding true materials are outside the desired range (Figure 2b). Examples are shown in Figure 3. 2) Greater conservatism leads to more materials in the desired range as shown by the fact that $\omega = 5$ outperforms $\omega = 1$ in all design cases. 3) Unconditioned policies manage to recreate the original distributions better than conditioned distributions. This is shown by better performance in pre-simulation metrics and in the plots in Figure 2c and Figure 2d, holding for both energy and band gap. 4) Random policies are not effective in generating valid and desired

| $\hat{p}$ | Example 1 | | Example 2 | |
|---|---|---|---|---|
| | True Crystal | Generated Crystal | True Crystal | Generated Crystal |
| 1.12 eV | $p = 2.157$ $Rb_2TlInBr_6$ | $p = 1.251$ $TlGaTe_2$ | $p = 3.280$ $Cs_2KYI_6$ | $p = 1.082$ $Cs2RbInI_6$ |
| 2 eV | $p = 1.049$ $TlGaTe_2$ | $p = 2.113$ $KAlTe_2$ | $p = 5.370$ $Mg_2P_2O_7$ | $p = 1.943$ $MgSnP_2O_7$ |
| 3 eV | $p = 3.959$ $Cs_2Si(HO_2)_2$ | $p = 3.243$ $Rb_2Si(HO_2)_2$ | $p = 1.150$ $Cs_2CuBiBr_6$ | $p = 2.977$ $K_2NaBiBr_6$ |
| 4 eV | $p = 5.097$ $Na_3InF_6$ | $p = 4.221$ $SrCaLa_4Zr_2O_{12}$ | $p = 0.801$ $TlIn_2GaTe_4$ | $p = 3.979$ $KNaCl_2$ |

Figure 3: Examples of cases where the crystal generated by our model has the band gap in the desired range i.e., $(\hat{p} - 0.25, \hat{p} + 0.25)$, while the ground truth crystal has the band gap outside the desire range. In most cases, it can be observed that some of the elements are common in the true and generated crystals. This indicates selective atomic substitutions for favorable band gap shift.

crystal structures. The average energy shown in Figure 4b for the random policy is lower than for the other policies, but this is not particularly meaningful as all policies manage to generate valid crystal structures. It is likely that the random policy generated a small subset of valid metal-like crystal structures given the close to zero average band gap shown in Figure 4a. Random policy generates many unrealistic crystals, since many of the DFT runs validating the crystals failed (Table 3), as well as the full experimental results showing pre-simulation metrics included in Table 1 and Table 2.

As shown in Figure 2, the higher values of $\hat{p}$ is more challenging because: 1) Most samples in the dataset have a lower band gap value (Appendix C) making the number of samples with a higher band gap that get exposed to the model while training a very small fraction, 2) Underestimation of band gaps by DFT, which causes an unfavorable shift from the expected band gap distribution.

## 6 CONCLUSION AND FUTURE WORK

We show that it is possible to train reinforcement learning based policies that can design valid crystal compositions conditioned on a crystal structure skeleton and a target property, such as the band gap, evaluated on precise and expensive computational chemistry engines, such as DFT. We demonstrate that offline RL methods can be used to learn distributions of design trajectories for valid crystal structures and provide tuning based on desired properties. While our results suggest that one can train policies for materials design problems, there is still significant space for future work to improve the performance, robustness and capabilities of the RL policies. First, our current approach only considers crystal structure composition, which can be extended to include additional design variables, such as crystal lattice parameters and atomic positions, for greater design flexibility to design more performant materials. Second, the dataset we used for offline RL is still limited in size ($\sim 10k$ materials) given the large cost of generating the dataset in a consistent manner and evaluating the reward function for structures generated by the policy. This leaves significant room for future work in creating large pretraining datasets and accelerating the evaluation of crystal structures through more optimized high-throughput DFT or machine learning based approximators. Third, much algorithmic work remains in designing better policies for materials design that can further improve the performance of conditional design.

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

# A EXPERIMENTAL DETAILS

## A.1 PRE-SIMULATION RESULTS

|  | Accuracy (%) | | Similarity (%) | | Validity (%) | |
|---|---|---|---|---|---|---|
| CQL Weight | $\omega = 1$ | $\omega = 5$ | $\omega = 1$ | $\omega = 5$ | $\omega = 1$ | $\omega = 5$ |
| *Random* | 0.0115 | | 0.1254 | | NaN | |
| *BC* | 52.26 | | 71.98 | | 85.00 | |
| *No Condition* | 49.77 | 51.53 | 70.85 | 71.26 | 81.50 | 82.54 |
| $(0 - 5 - 1)$ | 38.64 | 48.85 | 61.23 | 69.38 | 69.99 | 77.84 |
| $(0 - 5 - 3)$ | 43.02 | 46.43 | 65.01 | 67.04 | 73.57 | 78.44 |
| $(0 - 10 - 1)$ | 36.54 | 43.72 | 59.3 | 65.18 | 73.33 | 80.81 |
| $(0 - 10 - 3)$ | 35.16 | 42.42 | 57.48 | 64.15 | 71.20 | 81.30 |
| $(1 - 5 - 1)$ | 42.11 | 47.72 | 64.00 | 68.12 | 75.62 | 80.29 |
| $(1 - 5 - 3)$ | 40.59 | 47.57 | 63.70 | 67.26 | 72.93 | 76.51 |
| $(1 - 10 - 1)$ | 35.02 | 43.18 | 58.63 | 65.13 | 67.82 | 75.14 |
| $(1 - 10 - 3)$ | 35.38 | 43.81 | 57.23 | 65.58 | 61.87 | 77.19 |

Table 1: Pre-simulation metrics for band gap design case of 1.12 eV with $(\alpha_1 - \alpha_2 - \beta)$ corresponding to the terms of the reward function in Equation (7) with the policy in Figure 2 and best by metric highlighted. Unconditional policies perform better on pre-simulation metrics while conditioned policies produce target designs shown as in Figure 2 and discussed in Section 5.2.

|  | Accuracy (%) | | Similarity (%) | | Validity (%) | |
|---|---|---|---|---|---|---|
| CQL Weight | $\omega = 1$ | $\omega = 5$ | $\omega = 1$ | $\omega = 5$ | $\omega = 1$ | $\omega = 5$ |
| *Random* | 0.0115 | | 0.1254 | | NaN | |
| *BC* | 52.26 | | 71.98 | | 85.00 | |
| *No Condition* | 49.77 | 51.53 | 70.85 | 71.26 | 81.50 | 82.54 |
| $(0 - 5 - 1)$ | 41.82 | 48.09 | 64.34 | 68.82 | 80.21 | 82.18 |
| $(0 - 5 - 3)$ | 39.46 | 47.61 | 61.59 | 68.24 | 74.46 | 80.09 |
| $(0 - 10 - 1)$ | 33.24 | 39.42 | 60.78 | 53.42 | 62.39 | 67.82 |
| $(0 - 10 - 3)$ | 35.24 | 41.47 | 57.14 | 64.06 | 64.40 | 75.54 |
| $(1 - 5 - 1)$ | 38.80 | 46.79 | 60.09 | 68.77 | 70.80 | 80.17 |
| $(1 - 5 - 3)$ | 42.06 | 47.49 | 63.36 | 68.35 | 78.32 | 81.0 |
| $(1 - 10 - 1)$ | 36.52 | 42.21 | 59.57 | 65.07 | 76.55 | 74.41 |
| $(1 - 10 - 3)$ | 35.94 | 42.91 | 56.8 | 64.2 | 68.95 | 77.63 |

Table 2: Band gap design case of 4 eV with similar nomenclature and conclusions as Table 1.

The full algorithmic description as well as relevant hyperparameters related to the model architecture and policy training are shown below:

## A.2 MEGNET

- Number of MEGNet blocks: 3
- Node embedding dimensions: 16
- Edge embedding dimensions: 1
- State embedding dimensions: 8
- *READOUT* Function: Order-invariant *set2set* (Vinyals et al., 2015)

## A.3 OFFLINE RL

- Number of steps trained: 500000
- Discount factor: 0.99

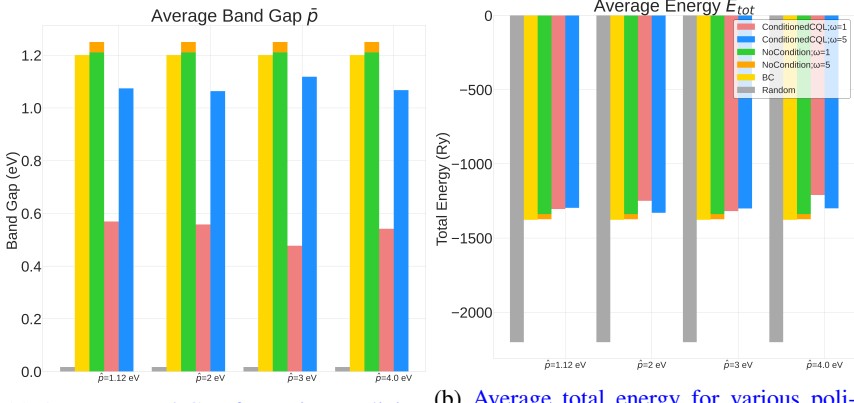

(a) Average Band Gap for various policies. Greater CQL conditioning ($\omega = 5$) yields greater alignment to the desired band gap for conditioned policies.

(b) Average total energy for various policies yielding valid crystals with energy below 0. Possible reasons for random policy having the lowest energies is provided in Section 5.2.

Figure 4: Analysis of average band gap and average energy of generated crystals in the validation set.

---

**Algorithm 1** Training Conditional CQL: DQN Version for Crystal Design with Target Property $\hat{p}$

---

Construct dataset $\mathcal{D}$ of size $N_{\mathcal{D}}$ consisting of transitions $(\boldsymbol{s}, \boldsymbol{a}, \boldsymbol{s}', r)$ using known crystals
Load $\mathcal{D}$ in Replay Buffer $\mathcal{B}$
Initialize Q-network $Q_\theta$ and target network $Q_{\theta'}$, batch size $B$
**for** $j = 1$ to max_steps **do**
    Sample $B$ transitions, $\{(\boldsymbol{s_i}, \boldsymbol{a_i}, \boldsymbol{s'_i}, r_i)\}_{i=1}^{B}$ from $\mathcal{B}$
    Compute TD loss
$$L_i^{TD}(\theta) = \begin{cases} (Q_\theta(\boldsymbol{s_i}, \boldsymbol{a_i}; \hat{p}) - (r_i + \gamma \max_{\boldsymbol{a}} Q_{\theta'}(\boldsymbol{s'_i}, \boldsymbol{a}; \hat{p})))^2 & \text{if } \boldsymbol{s'_i} \text{ is not terminal} \\ (Q_\theta(\boldsymbol{s_i}, \boldsymbol{a_i}; \hat{p}) - r_i)^2 & \text{otherwise} \end{cases}$$
    $L^{TD}(\theta) = \frac{1}{B} \sum_{i=1}^{B} L_i^{TD}(\theta)$
    Compute conservative loss, $L^C(\theta) = \frac{1}{B} \sum_{i=1}^{B} \left[ \log \sum_{\boldsymbol{a}} \exp(Q_\theta(\boldsymbol{s_i}, \boldsymbol{a}; \hat{p})) - Q_\theta(\boldsymbol{s_i}, \boldsymbol{a_i}; \hat{p}) \right]$
    Compute total CQL loss $L^{CQL}(\theta) = \omega L^C(\theta) + \frac{1}{2} L^{TD}(\theta)$
    Compute gradients and backpropogate: $\theta \leftarrow \theta - \eta \nabla L^{CQL}(\theta)$, $\eta$ is the learning rate
    Update target network parameters $\theta'$
**end for**

---

- Batch size: 1024
- Learning rate: $3e-4$

## A.4 DFT PARAMETERS (QUANTUM ESPRESSO)

For performing DFT calculations we use the Quantum Espresso (Giannozzi et al., 2009) simulation suite. The details of the DFT parameters are given below. For simplicity, this configuration was used for all crystals, and the evaluation is consistent for the training and generated crystals. Note that we do not perform structure relaxation in any of the cases.

- Calculation: SCF
- Tolerance: $1e-6$
- Number of Bands: 256
- $k$-points: (3-3-3)

- Occupations: fixed (since our training set consists only of nonmetallic crystals)
- Diagonalization: David
- ecutrho: 245
- ecutwfc: 30
- mixing_beta: 0.7
- degauss: 0.001
- Default charge: 0
- Maximum iterations: 1000

### A.4.1 HANDLING FAILURES

It is important to note that DFT can be best leveraged once we know certain properties of the crystals – for example, charge, magnetization, and metallicity. Considering the difficulty in determining these properties for completely unknown crystals, we standardized the evaluation procedure by using the same DFT configuration for all crystals (except for the crystal-specifc parameters like number of atoms, species, and pseudopotentials directory). However, this resulted in multiple crystals failing DFT simulation. Some of the errors are explained below.

- *Charge is wrong. Smearing is needed.*: This error mainly occurs because of unpaired electrons in the system, and can be resolved by changing the occupation to 'smearing' instead of 'fixed'. However, doing so will not help in determining the band gap of crystals, as it will only output the Fermi energy. Another way is to set the 'nspin' parameter to 2 and specify the total magnetization value as an additional input to Quantum Espresso. This helped us resolve most of the failures for the MP-20 crystals in the training and validation set because the total magnetization value is retrievable from the Materials Project, but for the newly generated crystals, we had to ignore those which failed because of this error. The error could also occur if generated crystal is metallic, and this property is also difficult to identify directly from the structure and composition.

- *NOT converged in 1000 iterations*: For some crystals, the DFT simulation did not converge even after 1000 iterations. These crystals were ignored while constructing the offline dataset, and also when evaluating the policy-generated crystals.

- *Time limit exceeded*: For constructing the offline dataset using known crystals, we used a flexible time limit to ensure none of the crystals are discarded because of time restrictions. However, while performing DFT simulation for the policy-generated crystals, due to the high-throughput nature of our evaluation pipeline, we had to ignore crystals that did not converge in 12 minutes.

- *Too few bands*: This error occurs when the number of bands specified, through 'nbnds' parameter is insufficient for the crystal system being simulated. This error was largely resolved by specifying a higher number of bands. In our case, we used 256 bands for all crystals.

Overall, during evaluation of generated crystals, only 50-70% of the valid crystals successfully underwent DFT simulation to output the energy and band gap (Table 3), and the rest failed because of the above errors.

### A.4.2 % DFT SUCCESS

Table 3 shows the percentage of policy-generated crystals that successfully underwent DFT simulation based on failure handling strategies discussed in Appendix A.3.

## B LIMITATIONS

The important limitations of this work are that the scope is limited to predicting only the atom types given all other information about the skeleton of the crystal and the order of traversal, and the

| | % DFT Success | |
|---|---|---|
| **CQL Weight** | $\omega = 1$ | $\omega = 5$ |
| *Random* | 14.99 | |
| *BC* | 67.48 | |
| *No Condition* | 70.97 | 59.92 |
| CQL($\hat{p} = 1.12$ eV) | 51.92 | 61.68 |
| CQL($\hat{p} = 2$ eV) | 53.76 | 69.06 |
| CQL($\hat{p} = 3$ eV) | 54.18 | 67.71 |
| CQL($\hat{p} = 4$ eV) | 52.31 | 66.48 |

Table 3: % Generated valid crystals that successfully underwent DFT simulation, for random policy and each of the trained models. Most of the crystals generated by the random policy failed DFT simulation.

training data is small and limited to nonmetals. Considering computational challenges attributed to DFT calculations, we had to restrict our design parameter space to a very small set, but it would be interesting to see the results after an extensive analysis after training models with several values of $\omega, \alpha_1, \alpha_2$, and $\beta$. Due to significant underestimation of band gaps by DFT, most of the generated crystals had an estimated band gap value of 0.0, which hindered our evaluation and analyses. This explains the very low fraction of generated crystals having a greater band gap.

## C TRUE DISTRIBUTIONS OF PROPERTIES

This section shows the true distribution of the band gaps and total energies for both training and validation data.

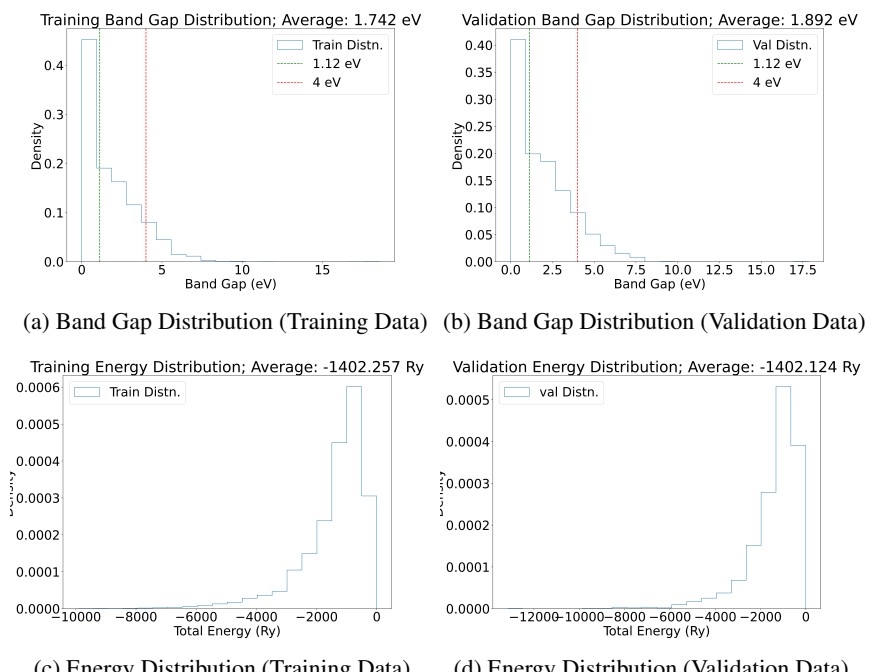

(a) Band Gap Distribution (Training Data)  (b) Band Gap Distribution (Validation Data)

(c) Energy Distribution (Training Data)  (d) Energy Distribution (Validation Data)

## D FULL EXPERIMENTAL POST-SIMULATION METRICS

We provide full experimental for our reward function design parameters for both the 1.12 eV design case (Table 1 and Figure 6 and 4 eV case (Table 2 and Figure 7) below. The tables includes evaluation of both the pre-simulation and post-simulation metrics described in Section 5.

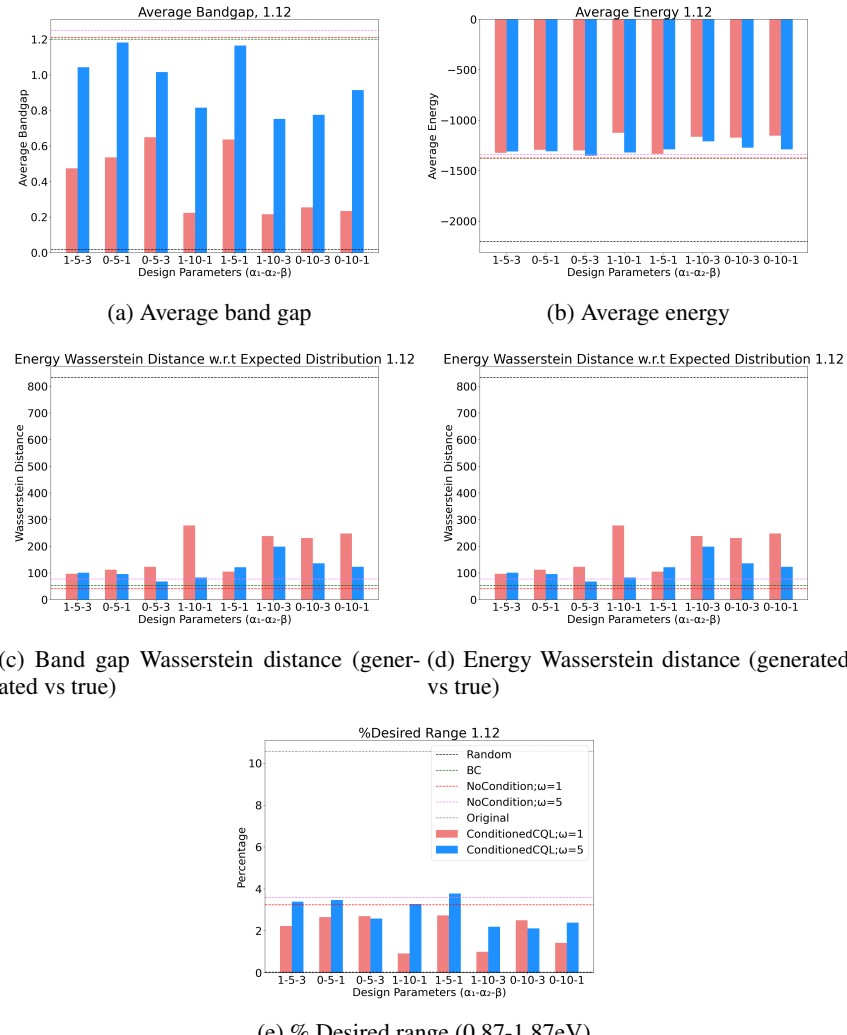

(a) Average band gap

(b) Average energy

(c) Band gap Wasserstein distance (generated vs true)

(d) Energy Wasserstein distance (generated vs true)

(e) % Desired range (0.87-1.87eV)

Figure 6: Full design parameter values for all learned policies for the band gap design case of 1.12 eV. Nomenclature of the table is $(\alpha_1 - \alpha_2 - \beta)$ corresponding to the terms of the reward function in Equation (7)

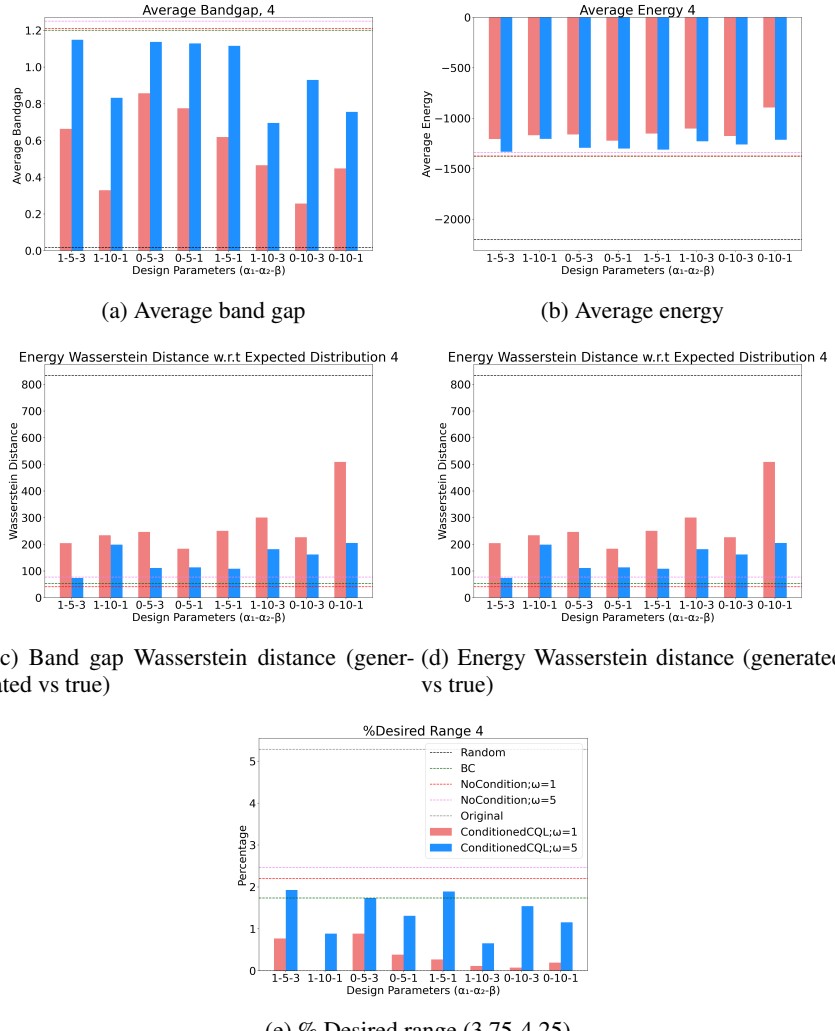

(a) Average band gap

(b) Average energy

(c) Band gap Wasserstein distance (generated vs true)

(d) Energy Wasserstein distance (generated vs true)

(e) % Desired range (3.75-4.25)

Figure 7: Full design parameter values for all learned policies for the band gap design case of 4.0 eV. Nomenclature of the table is $(\alpha_1 - \alpha_2 - \beta)$ corresponding to the terms of the reward function in Equation (7)

