# OpenReview forum: "Learning Conditional Policy for Crystal Design using Offline Reinforcement Learning"
_ICLR.cc/2024/Conference — ICLR 2024 Conference Withdrawn Submission_

### Official Review · Reviewer_NogB · 2023-10-30

**Soundness:** 3 good
**Presentation:** 3 good
**Contribution:** 2 fair
**Rating:** 5
**Confidence:** 2

**Summary:**

The paper tries to design novel and stable crystalline materials conditioned on a desired band gap. The authors turn the problem into a sequential decision-making problem and use offline RL algorithms to learn an effective policy to design materials. They design the reward function for material design. For experiments, they compare their methods against behavior cloning, random policy, etc, and demonstrate the effectiveness.

**Strengths:**

+ They formulate the crystal design problem into an offline RL problem which utilizes total energy and band gap values to get reward functions. By using DFT calculations, they can optimize the rewards thus solving the crystal structure problems;
+ They built an offline dataset from the Materials Project so that offline RL algorithms could learn from the dataset;
+ The illustration and explanation of the material designs are easy to understand for ML people, the authors abstract away the complex details and make an effort in the presentation to make the paper accessible.

**Weaknesses:**

- I am in no way a crystal design expert, but from the experiments, it feels like lacking strong baselines.  The authors compare their method to some natural baselines deviated from their proposed method (CQL). It would be great if the authors could compare/mention the state-of-the-art methods used in the field of crystal design;

**Questions:**

The expert reviewer on DFT/crystal designs should do their gatekeeping job on the paper. From my educated guess from an RL researcher's point of view, the paper is ok.

---

> ### Author Response · Authors · 2023-11-21
> **Author's Response**
>
> We would like to thank you for your review, and for your constructive comments about the paper. We truly appreciate your points on the clarity of the paper. Please find the explanations for your comments below.
> ___
> > *From the experiments, it feels like lacking strong baselines*
>
> For our experiments, we primarily focused on **unconditional baselines (random, behavioral cloning, and unconditioned CQL)** for a better understanding of whether conditioning helps in improving the property of interest. As suggested by **Reviewer ffhH**, it would also be interesting to compare the performance with conditional baselines with approaches like behavioral cloning (supervised training on materials that have the target property). We are currently working on this aspect.
> ___
> > *It would be great if the authors could compare/mention the state-of-the-art methods used in the field of crystal design*
>
> Diffusion models[1], GANs[2][3], and RL[4] have been used to address the crystal design problem. Other approaches are cited in **Section 2** in the paper. However, as stated in our response to **Reviewer b3qf**, there are various reasons why a direct comparison of our approach with those models is difficult.  An important reason is that most approaches that use DFT for evaluation focus only on a small chemical space or have shape constraints (e.g. cubic). Further, our way of formulating the problem as a sequential decision-making process to predict atom types is entirely different from existing approaches to material design. We do not predict the structure, making it difficult to compare our approach with models that generate the structure. Lastly, none of the previous studies focused on band gap directed inverse design of crystals.
> ___
> **We wish to thank the reviewer for the positive ratings and feedback. If you are satisfied with our response and the updated draft, it would be great if there is any scope for improvement of your rating. If you have more questions, we would be happy to answer them.**
> ___
> **References**
>
> [1] Xie, Tian, et al. "Crystal diffusion variational autoencoder for periodic material generation." arXiv preprint arXiv:2110.06197 (2021).
>
> [2] Zhao, Yong, et al. "Physics guided deep learning for generative design of crystal materials with symmetry constraints." npj Computational Materials 9.1 (2023): 38.
>
> [3] Zhao, Yong, et al. "High‐throughput discovery of novel cubic crystal materials using deep generative neural networks." Advanced Science 8.20 (2021): 2100566.
>
> [4] Pan, Elton, Christopher Karpovich, and Elsa Olivetti. "Deep reinforcement learning for inverse inorganic materials design." arXiv preprint arXiv:2210.11931 (2022).

---

> > ### Comment · Reviewer_NogB · 2023-11-22
> >
> > As I said, I am in no way an expert in material design. After reading my fellow reviewers' comments (especially from the reviewer ffhH) and the authors' rebuttal, I think the paper does not qualify as a good scientific material design application of offline RL algorithms. Thus, I have lowered my score.

---

### Official Review · Reviewer_W7dF · 2023-10-31

**Soundness:** 3 good
**Presentation:** 3 good
**Contribution:** 2 fair
**Rating:** 5
**Confidence:** 2

**Summary:**

This work focuses on inverse design for crystal structures which has a desired band gap. They model the problem as a MDP, and apply off-the-shelf algorithms from offline RL.

The results are evaluated by DFT, therefore, might be promising. But I'm not an expert on band gap, so I cannot judge whether the results are strong.

**Strengths:**

RL-based inverse design for 3D structures is relatively under-explored. This work proves that RL methods have the potential to resolve some challenges of inverse design in science.

**Weaknesses:**

1. No novel algorithm is proposed. But for application-oriented research, novel algorithm is not always necessary. The focus here is on implementing and adapting existing methods in a novel context.
2. The analyses of the results are limited.  It would be beneficial to quantify how many new structures were generated and clarify whether the algorithm simply reproduces known structures from the training set. Given that conservative RL is used, I suspect the output is close to the training set.
3. Since DFT is actually included in the loop, how about using online RL? Is there any chance to find some novel materials through automated exploration.

**Questions:**

See above.

---

> ### Author Response · Authors · 2023-11-21
> **Author's Response**
>
> We wish to thank you for your review and acknowledgement of our contribution to the underexplored avenue of RL-based inverse design of materials. We have provided explanations for your comments and concerns below.
> ___
> > *No novel algorithm is proposed. But for application-oriented research, novel algorithm is not always necessary. The focus here is on implementing and adapting existing methods in a novel context.*
>
> We acknowledge and agree with your comment. The primary area of our submission is **“applications to physical sciences (physics, chemistry, biology, etc.)”**, one of the topics in ICLR 2024 (https://iclr.cc/Conferences/2024/CallForPapers), and while our approach relies on the CQL approach, the construction of our DFT-integrated workflow is unique and novel to the best of our knowledge.
> ___
> > *The analyses of the results are limited. It would be beneficial to quantify how many new structures were generated and clarify whether the algorithm simply reproduces known structures from the training set. Given that conservative RL is used, I suspect the output is close to the training set.*
>
> The pre-simulation metrics in **Section 5.1** of the paper indicate the accuracy and similarity (substitutions with similar atoms) of the trained models on the validation data. While the accuracy, which measures the percentage of accurately predicted atoms globally, was between **36-50%** for all models on the validation dataset, the fraction of exactly reconstructed crystals (i.e., all atoms match the ground truth in a crystal) was extremely low (0-7% depending on the model). This could be due to two reasons: 1) the optimization objective of CQL[1] aims not just to fit the dataset (as in BC), but to balance the tradeoff between minimizing the Bellman error and being conservative, 2) the offline policy is trained with batches of transitions i.e., $(s, a, s^{\prime}, r)$, but during test time we perform a rollout starting from an initial state $\boldsymbol{s_0}$. This makes it harder for the learned policy to exactly reconstruct the crystals. Nevertheless, this contributes to the novelty of the generated crystals – more than 90% of the generated crystals of all models had a different composition compared to the corresponding ground truth crystals. **Figure 3** in our updated draft shows examples of generated crystals that have the desired property of interest when the corresponding true crystals lie outside the desired range. While the model does reconstruct the crystals in the training set more accurately than the validation set, the points discussed above hold for the training set, too, and this prevents it from reaching the maximum accuracy (100%). About your comment on the extent of the analysis, we have added results for more design targets (2 eV and 3 eV) and added a new metric **(out-of-distribution design)** in our latest draft of the paper (Figure 2).
> ___
> > *Since DFT is actually included in the loop, how about using online RL? Is there any chance to find some novel materials through automated exploration?*
>
> Directly using reward signals from DFT calculators in an online reinforcement learning setup is extremely difficult to train because of the large computation time of DFT simulations. However, we believe that the learned offline policy could serve as a good initial policy (thereby incorporating inductive bias) for online RL, and this would be the immediate future work. For online experiments, it is more reasonable to start with a single or a small batch of crystal skeletons in a way that the training process does not take a very long time.
> ___
>
> **We would like to thank the reviewer for the constructive feedback and positive rating. We hope our responses answered your queries, and are happy to answer more if required. If you are satisfied with the replies and the updated draft, it would be great if there is any scope for improvement in your score for our submission.**
> ___
> **References**
>
> [1] Kumar, Aviral, et al. "Conservative q-learning for offline reinforcement learning." Advances in Neural Information Processing Systems 33 (2020): 1179-1191.

---

### Official Review · Reviewer_b3qf · 2023-10-31

**Soundness:** 2 fair
**Presentation:** 2 fair
**Contribution:** 1 poor
**Rating:** 3
**Confidence:** 5

**Summary:**

The authors formalize the problem of crystal design as a MDP, produce an offline dataset for this MDP, and train CQL policies conditioned on the band gap desired in order to attempt to design novel semiconductors. They evaluate a conditional CQL policy along side a random policy, behavior cloning, and an unconditional CQL policy.

They evaluate whether the crystal is valid, whether the total energy is low enough, and whether the generated and true band-gap distributions are accurate.

They find that conditional policies generate materials closer to the desired band gap range, that greater conditioning seems to lead to better outputs, that unconditional policies recreate the original distirbution of data better, and that random policies do not generate good output.

**Strengths:**

Crystal design is an important problem for society and it is important that we bring our best tools to bear. Determining whether ML can be helpful and how best to apply it is a useful scientific enterprise and it seems this paper is a step in that direction. It seems that the authors have created 1) a useful new method, 2) a dataset that might be of independent value to the matsci community. These are both valuable.

**Weaknesses:**

This paper is not suitable for the program of ICLR, as it doesn’t advance the state of the art in machine learning. CQL is at this point a widely known algorithm and the application to crystal design is not one which pushes ML forward.

It is also not clear to me that the MDP formalism is the correct one for this problem setting. It seems to me that using a conditional generative model over the entire data would be a more natural choice (think diffusion model or autoregressive transformer). The authors don’t evaluate against those baselines which I think might be better suited.

**Questions:**

Have people tried generative modeling approaches to this problem? How well did they work? Can we quantitatively compare?

Is there a venue in materials science that would be more appropriate for this work?

---

> ### Author Response · Authors · 2023-11-21
> **Author's Response**
>
> We thank you for the review and feedback, and for acknowledging the importance of the crystal design problem. We appreciate your comment on the usefulness of our method to the material science community. Please find our comments for your questions and concerns below.
> ___
> > *This paper is not suitable for the program of ICLR, as it doesn’t advance the state of the art in machine learning.*
>
> We wish to clarify that the primary area of our submission is **“applications to physical sciences (physics, chemistry, biology, etc.)”**, which is mentioned as one of the topics/tracks in ICLR 2024’s call for papers (https://iclr.cc/Conferences/2024/CallForPapers) and therefore represents an important area for ICLR.
> ___
> > *CQL is at this point a widely known algorithm and the application to crystal design is not one which pushes ML forward.*
>
> We acknowledge that our approach relies on the CQL approach for the offline training. However, to the best of our knowledge, our crystal design pipeline is novel, where we have leveraged large datasets and **first-principles DFT calculations**, and integrated them into the offline RL training process – while DFT helps in accurate estimation of properties like energy and band gap, the cost of computation is very high. It takes a few seconds to several minutes (~2 minutes on average) to compute the reward for each input crystal. Through our approach, we demonstrate promising results for band-gap-directed crystal design, highlight the challenges and scope for future work, and propose to release a benchmark dataset for the benefit of the material discovery community. We believe that this is a solid contribution to the **“applications to physical sciences”** track in ICLR.
> ___
> > *It is also not clear to me that the MDP formalism is the correct one for this problem setting. It seems to me that using a conditional generative model over the entire data would be a more natural choice (think diffusion model or autoregressive transformer). The authors don’t evaluate against those baselines which I think might be better suited.*
>
> Diffusion models have previously been studied for material design[1]. However, this involves predicting the identities/properties of all atoms at once with no sequential process involved. The case is the same for other generative models like GAN. We believe that by framing this problem as a sequential decision-making process, the important aspects of **exploration, credit assignment, and ‘learning to search solutions’ are more intuitive**, especially when integrated with DFT – this could help in improving the explainability of the model. MDP formalism has proven very useful in small molecule discovery (drug design)[5][6], and is also of interest to the material discovery community[7]. We primarily focused on unconditional baselines (random, behavioral cloning, and unconditioned CQL) in our experiments for an understanding of whether conditioning helps in improving the property of interest. We are currently working on evaluating diffusion and decision transformer-based conditional models for band gap directed crystal design.
> ___
> > *Have people tried generative modeling approaches to this problem? How well did they work? Can we quantitatively compare?*
>
> CDVAE[1] is the first approach to address the crystal structure design problem using a diffusion model. However, there is no integration of first-principles DFT properties, and only approximate metrics and properties were used for performance evaluation. Other works focused on generative models like GANs, with the most recent one being [2], where generated crystals were evaluated using DFT. Another work is [3], which used a distributional graphormer model to generate carbon-based crystal structures. While the performances of these methods are promising, the reasons why we cannot perform a direct comparison of these models with our approach are mentioned below.
> - Many approaches that use DFT focus only on small chemical systems, with lesser number of elements in the vocabulary, or have constraints on the shape of crystals. For instance, Distributional graphormer[3] generates only carbon-based crystal structures. CubicGAN[4] focuses only on cubic crystals. However, a conditional variant of the GAN model proposed by [2] could be more suitable for comparison with our model in terms of property-directed generation. We are working on the same.
> - Some of the metrics used by structure generation models (e.g., structural validity as in [1]) cannot be directly used for comparison with our model, since we do not generate the structure.
> - None of the previous studies focused on inverse design of crystals (without structural or composition constraints) with band gap as the property of interest.

---

> > ### Author Response · Authors · 2023-11-21
> > **Author's Response (Contd.)**
> >
> > > *Is there a venue in materials science that would be more appropriate for this work?*
> >
> > As previously stated, we focus on ICLR’s track on “applications to physical sciences (physics, chemistry, biology, etc.)” as our primary area, and we believe that our work is a fruitful contribution to advancing the machine learning pipeline for material discovery.
> > ___
> > **We thank the reviewer once again for the feedback and questions. We are happy to answer any more questions. If your major concerns have been addressed, and given the suitability of our submission to the ‘applications to physical sciences’ topic in ICLR 2024, we kindly request you to consider revising your review.**
> > ___
> > **References**
> >
> > [1] Xie, Tian, et al. "Crystal diffusion variational autoencoder for periodic material generation." arXiv preprint arXiv:2110.06197 (2021).
> >
> > [2] Zhao, Yong, et al. "Physics guided deep learning for generative design of crystal materials with symmetry constraints." npj Computational Materials 9.1 (2023): 38.
> >
> > [3] Zheng, Shuxin, et al. "Towards Predicting Equilibrium Distributions for Molecular Systems with Deep Learning." arXiv preprint arXiv:2306.05445 (2023).
> >
> > [4] Zhao, Yong, et al. "High‐throughput discovery of novel cubic crystal materials using deep generative neural networks." Advanced Science 8.20 (2021): 2100566.
> >
> > [5] Ghugare, Raj, et al. "Searching for High-Value Molecules Using Reinforcement Learning and Transformers." arXiv preprint arXiv:2310.02902 (2023).
> >
> > [6] Korshunova, Maria, et al. "Generative and reinforcement learning approaches for the automated de novo design of bioactive compounds." Communications Chemistry 5.1 (2022): 129.
> >
> > [7] Pan, Elton, Christopher Karpovich, and Elsa Olivetti. "Deep reinforcement learning for inverse inorganic materials design." arXiv preprint arXiv:2210.11931 (2022).

---

### Official Review · Reviewer_ffhH · 2023-11-01

**Soundness:** 2 fair
**Presentation:** 2 fair
**Contribution:** 2 fair
**Rating:** 3
**Confidence:** 4

**Summary:**

This paper tackles the problem of inversely designing crystals by using offline reinforcement learning on a materials dataset with recomputed DFT property labels. The goal of this work is to learn a conditional policy that can efficiently perform atom substitution of given crystal skeletons, such that the resulting material satisfies a given target condition. It is shown that the learned conditional policy outperforms several baselines on a crystal design task where the generated materials have to satisfy particular band gap constraints.

**Strengths:**

The paper is clearly written and the use of offline RL is well-motivated by the prohibitively expense DFT calculations that would be required in an online setting. To my knowledge, this is also the first paper that uses offline RL for materials design. Limitations of the approach are clearly communicated, too.

Overall, the chosen approach makes sense and the definition of the reward function and the Q-function are reasonable. Having DFT energies and band gap values consistently calculated with QE could be useful for researchers that cannot afford commercial DFT software.

**Weaknesses:**

As the authors point out, the paper addresses a vastly simplified problem. The policy has a very limited action space, and is basically just learning a more efficient way to substitute atoms. I do not see any evidence that the approach could be easily extended/scaled to a more complex action space as described in the future work section, so I am not sure how useful this approach is in practice. The RL approach itself is not very original since it is mostly a straightforward adaptation of CQL. And while I appreciate the effort to generate the dataset with QE, I would have liked to see more details about how the dataset was generated (e.g., which settings have been used and how were failures handled) if this is one of the main contributions of the paper.

The paper also has substantial flaws in its methodology. First of all, total energy says nothing about thermodynamic stability; instead, energy above convex hull is a much better measure. Secondly, I found the presented baselines pretty weak since they are either random or completely oblivious of the target condition (if I understand correctly). For example, a stronger BC baseline might be to train the policy on only those materials whose band gap values are close to the target condition. Finally, I would have liked to see how the generated crystals look like.

In terms of presentation, I found the quality of the tables and figures to be quite low. Given that this is an RL for materials design paper, I also would have liked to see a few more references at this intersection. See below for a few examples (however, note that I have not checked all of them thoroughly):
Sui et al. (2021). Deep Reinforcement Learning for Digital Materials Design.
Law et al. (2022). Upper-Bound Energy Minimization to Search for Stable Functional Materials with Graph Neural Networks.
Pan et al. (2022). Deep Reinforcement Learning for Inverse Inorganic Materials Design.
Zheng et al. (2022). Designing mechanically tough graphene oxide materials using deep reinforcement learning.

Minor comments:
- p. 1: "large and discrete space" - While the search space in the paper is discrete, the space of all possible materials is not discrete.
- p. 2: "in the Materials Project database" - citation missing
- p. 3: "is the is the" -> "is the"
- p. 3: "230 having the" -> "230 has the"
- Table 2: Given that the table does not add much additional information, I would suggest to move this to the appendix

**Questions:**

1. Could you please provide more information regarding how the dataset was generated?
2. How is the initial crystal skeleton $G_0$ for each episode chosen/sampled?
3. How do the crystals generated by the policy look like? Are they sensible?

---

> ### Author Response · Authors · 2023-11-21
> **Author's Response**
>
> Thank you for the insightful analysis of the paper. The strengths you mentioned about the reasonableness and usefulness of our methodology and the clarity of the paper are very encouraging. We also appreciate the domain-related comments in your review. Please find our answers to your questions and concerns below.
> ___
> > *The policy has a very limited action space, and is basically just learning a more efficient way to substitute atoms.*
>
> While this problem is a simplified version of the overall goal of full crystal structure design, we wish to point out that our action space comprises **88 elements** in the periodic table (the action space is larger than most of the common RL benchmarks like Atari), and the mapping between the crystal structure and DFT properties is quite complex for any model to learn, which is why we consider this as a ‘learning to search’ problem. Our main hypothesis is the following – among the combinatorial possibilities of atom substitutions given a skeleton of a crystal, there exist some substitutions that result in better stability and have the desired property of interest, and the RL model is expected to search for these solutions. Further, the challenges such as less exploration in offline RL and underestimation of DFT-based band gaps, pose additional difficulties. Also, through this work, we aim to accelerate **high-throughput virtual screening (HTVS)**, which commonly substitutes atoms in existing crystal structures to determine novel crystals – this has resulted in augmentations of popular crystal structure databases such as the **Materials Project**[1].
> ___
> > *I do not see any evidence that the approach could be easily extended/scaled to a more complex action space as described in the future work section, so I am not sure how useful this approach is in practice.*
>
> Our future work includes extending the action space to continuous variables like the position of atoms and the lattice structure, in addition to the discrete atom types. This would involve modeling discrete and continuous policies together (or handling multiple action types), which is currently an interesting problem in the RL community. Based on the promising results of previous works that have attempted to tackle this multi-dimensional complex action space, we believe this to be feasible for future work [2][3]. Building on these works for learning policies for atom type and structure generation is a useful extension of our contribution, which addresses the overall problem of designing crystals without prior information about the structure. The inherent challenges of this are yet to be investigated, but we believe that our DFT-integrated offline RL pipeline and evaluation strategies would be very helpful for this challenging design setting.
> ___
> > *I would have liked to see more details about how the dataset was generated (e.g., which settings have been used and how were failures handled) if this is one of the main contributions of the paper*
>
> Thanks for the suggestion. We have updated our submission with **Section A.4** in the Appendix, which explains the details of the DFT computations and handling of failures.  As explained in **Section 4.4**, we construct state trajectories using crystals in the Materials Project dataset; an example of a trajectory is shown in **Figure 1a** in the paper. Further details are provided in our response to your question on how the dataset was generated.
> ___
> > *Total energy says nothing about thermodynamic stability; instead, energy above convex hull is a much better measure.*
>
> At present, through our unique reward scheme, we only aim to design crystals with desired band gap that are generally considered stable so they can be used for practical purposes. This is also why the coefficient of the energy term in our reward formulation ($\alpha_1$) is lower than that of the property distance term ($\alpha_2$). However, we understand and acknowledge your concern about total energy. We are constantly looking for reward schemes that better capture the energetic properties of the crystals in a relative sense rather than absolute. While the energy above the convex hull is the best-known measure of a crystal’s stability [4], this metric is conditioned on the elements present in the crystal and is computationally expensive to calculate for a large number of crystals with varying compositions. We are currently redoing our analysis with **formation energy**, which is not as expensive as energy above convex hull but a more reasonable metric to incorporate in the reward function since it represents a better approximation for crystal stability.

---

> > ### Author Response · Authors · 2023-11-21
> > **Author's Response (Contd.)**
> >
> > > *I found the presented baselines pretty weak since they are either random or completely oblivious of the target condition (if I understand correctly). For example, a stronger BC baseline might be to train the policy on only those materials whose band gap values are close to the target condition*
> >
> > Thank you for this feedback about including target conditions in our baselines. In the coming days, we will be adding the results corresponding to BC trained on materials with band gaps close to the target.
> > ___
> > > *I would have liked to see how the generated crystals look like.*
> >
> > We have added a figure (Figure 3 in the main paper, under Section 5.2) containing some examples of generated crystals for each target band gap. We have selected examples from the validation set where the true crystal does not have the band gap in the desired range, but the corresponding policy-generated crystal has the desired band gap. The visualization was performed with VESTA[5]. The structures of the generated crystal and the corresponding true crystal would be the same in our case. Only the composition will differ. The figure suggests that in multiple cases, the policy generated a composition that is conducive to the target band gap.
> > ___
> > > *I found the quality of the tables and figures to be quite low.*
> >
> > We have updated our submission with the following changes in the images and tables.
> > - Added **Figure 3** in Section 5.2 for visualizing generated crystals.
> > - Improved quality of tables and moved Table 1 and Table 2 to the appendix to make space for Figure 3.
> > - Figure 2 is updated with results from **two more band gap targets (2 eV and 3 eV)**. Figure 3 includes these new targets.
> > - The desired range plot (**Figure 2a**) is updated after some corrections
> > - A new metric is introduced – **Out-of-distribution (OOD) design**, shown in **Figure 2b**. It measures the ability of the policy to generate crystals with the desired band gap when the ground truth band gaps do not fall in the desired range.
> > - Moved average band gap and average energy plot to the Appendix (**Figure 4**).
> > ___
> > > *Minor comments (typos, citation missing, comment on discrete chemical space, table)*
> >
> > All points mentioned in the 'Minor comments' section in the review have been addressed in the updated draft of our submission.

---

> > > ### Author Response · Authors · 2023-11-21
> > > **Author's Response to Questions**
> > >
> > > > *Could you please provide more information regarding how the dataset was generated?*
> > >
> > > The offline dataset generation process (Section 4.4 in the paper) works as follows. Given a set of crystal structures from the Materials Project,
> > > - Step 1: Compute the energies ($E_{tot}$) and band gaps ($p$) (and, therefore, rewards) by performing DFT simulation using Quantum Espresso – details of DFT settings are given in **Section A.4** in the Appendix.
> > > - Step 2: Construct episodic state trajectories $(s,a,s^\prime,r)$, as in **Figure 1a** by representing each state as a graph with atom types as node features. The initial state is a graph with all atom types hidden (using a special token in one-hot encoding). The state also carries information about which node to **‘focus’** for atom type prediction. In our case, intermediate rewards are zero, and the terminal reward is a function of energy and band gap (according to **Equation 7**)
> > > - Step 3: Multiple trajectories are constructed for each crystal with different start nodes (the start node is indicated by the 'focus' variable in the initial state graph), and the order of traversal is determined by breadth-first search (BFS) traversal.
> > > - Step 4: The transitions are stored in the replay buffer prior to training the policy.
> > > ___
> > > > *How is the initial crystal skeleton for each episode chosen/sampled?*
> > >
> > > During training, state transitions $(s,a,s^\prime,r)$ are uniformly sampled in batches for learning the offline policy. During test time, we start with the initial crystal skeleton of the crystal we want to complete, and perform a rollout using the learned offline policy $\pi_o(\boldsymbol{a}|\boldsymbol{s}, \hat{p})$ to obtain the completed crystal structure. In our experiments, we perform rollouts for **all the crystals in the validation set** during test time.
> > > ___
> > > > *How do the crystals generated by the policy look like? Are they sensible?*
> > >
> > > At present, we measure only the compositional validity of the generated crystals, similar to CDVAE[6], which is reported in our submission in Tables 1 and 2. Figure 3 in our updated draft shows some examples of the true and generated crystals for each band gap target. We do not optimize for the structure or perform any relaxation, and hence, the **structures of the true and generated crystals are the same**. However, the compositions might differ, and in multiple instances, the newly generated crystal composition leads to a band gap value closer to the target (Figure 3 and Figure 2d). Therefore, they are sensible.
> > >
> > > ___
> > > **To conclude, we thank the reviewer once again for the useful and domain-related comments, and for the detailed analysis. We are happy to answer any more questions if required or incorporate other suggestions for our final draft. If your questions and concerns were adequately addressed in our rebuttal, we humbly request your consideration in enhancing the score for our submission.**
> > > ___
> > >
> > > **References**
> > >
> > > [1] Jain, Anubhav, et al. "Commentary: The Materials Project: A materials genome approach to accelerating materials innovation." APL materials 1.1 (2013).
> > >
> > > [2] Delalleau, Olivier, et al. "Discrete and continuous action representation for practical rl in video games." arXiv preprint arXiv:1912.11077 (2019).
> > >
> > > [3] TAC: Towered Actor Critic for Handling Multiple Action Types in Reinforcement Learning for Drug Discovery
> > >
> > > [4] Bartel, Christopher J., et al. "A critical examination of compound stability predictions from machine-learned formation energies." npj computational materials 6.1 (2020): 97.
> > >
> > > [5] Momma, Koichi, and Fujio Izumi. "VESTA 3 for three-dimensional visualization of crystal, volumetric and morphology data." Journal of applied crystallography 44.6 (2011): 1272-1276.
> > >
> > > [6] Xie, Tian, et al. "Crystal diffusion variational autoencoder for periodic material generation." arXiv preprint arXiv:2110.06197 (2021).

---

> ### Comment · Reviewer_ffhH · 2023-11-22
>
> Thank you for your reply.
>
> I agree that substitution is a widely used approach and a more efficient way to do that might be valuable, at least for systems with four or more distinct atom types. My main concern with the future work is not that it is difficult to model, but that it will make the problem significantly harder. In particular, it is much more complex than molecular design since the number of elements is much larger, there are much more degrees of freedom, and there are complex interactions with the lattice.
>
> I disagree that energy above hull is too expensive to compute, but I think formation energy would also be acceptable given that this is an ML conference, so I appreciate your efforts. If you really cannot afford to compute energy above hull for every generated structure, I believe a reasonable compromise would be to use formation energy for the reward but additionally compute energy above hull only on the crystals generated via the validation set.
> If you could show that your approach still works in this case, I would be willing to revise my score from 3 to 5. However, as it stands I will maintain my current score.

---

### Comment · Area_Chair_LXEr · 2023-11-21
**To all reviewers: Please respond to the authors' rebuttal**

Dear reviewers,

The authors have just submitted their rebuttal. Given that the window for interacting with authors on their rebuttal is closing soon (on Wednesday Nov 21st), please respond to the authors' rebuttal as soon as possible, so that you can discuss any agreements or disagreements. Please acknowledge that you have read the authors' comments, and explain why their rebuttal does or does not change your opinion and score.

Many thanks,

Your AC